# Bioinformatics Predicted Linear Epitopes of the Major Coat Protein of the Beet Yellows Virus for Detection of the Virus in the Cell Extract of the Infected Plant

**DOI:** 10.3390/biotech11040052

**Published:** 2022-11-10

**Authors:** Eugene V. Skurat, Konstantin O. Butenko, Yuri F. Drygin

**Affiliations:** 1Faculty of Biology, Lomonosov Moscow State University, 119991 Moscow, Russia; 2Belozersky Research Institute of Physico-Chemical Biology, Lomonosov Moscow State University, 119991 Moscow, Russia

**Keywords:** bioinformatics, recombinant DNA, fused protein, 3D protein structure, Ni-NTA chromatography, antiserum, ELISA

## Abstract

Beet yellows virus, which belongs to the genus *Closterovirus*, family *Closteroviridae* and has a significant negative economic impact, has proven to be challenging to detect and diagnose. To obtain antibodies against BYV, we propose an easier bioinformatics approach than the isolation and purification of the wild virus as an antigen. We used the SWISS-MODEL Workspace (Biozentrum Basel) protein 3D prediction program to discover epitopes of major coat protein p22 lying on the surface of the BYV capsid. Sequences coding these epitopes were cloned into plasmid pQE-40 (Qiagen) in frame with mouse dihydrofolate reductase gene. Fused epitopes were expressed in *Escherichia coli* and isolated by the Ni-NTA affinity chromatography. Murine antibodies were raised against each epitope and in a combination of both and characterized by dot-ELISA and indirect ELISA. We successively used these antibodies for diagnosis of virus disease in systemically infected *Tetragonia tetragonioides*. We believe the approach described above can be used for diagnostics of difficult-to-obtain and hazardous-to-health viral infections.

## 1. Introduction

Beet yellows virus (BYV) has a filamentous and exceptionally long capsid of 1350 nm in length. Viral particles contain a single strand, up to 15,480 nucleotides long, RNA genome, which is packed with two capsid proteins (CP): minor p24, which packs 5% of the genome and major p22, which packs the rest of the genome [1,2].

BYV has significant impact on sugar beets crops. In nature, the virus is usually transmitted by multiple species of aphid. BYV infection can cause 1.5–2.7% reduction in sugar content and reduce crops by 25–65%, especially if the beets are infected at the early stage of the development. As plant viral infections are impossible to cure at this time, the main approach of avoiding viral infections is quarantine measures and mass diagnostic of the infection on a molecular level.

Traditionally in horticulture, mass diagnostics are performed using immunochemistry. In cases of viroid or difficult to isolate viruses, the mass diagnostics relies instead on nucleic acids’ hybridization [3]. Technical and biosafety challenges of infecting plants using aphids or through the viral inoculation combined with low viral yield (<10 mg of virus per 100 g of infected leaves) significantly complicate the isolation of the amount of the pure virus that is necessary for immunization. This is further impacted by low stability of BYV viral particles due to the degradation by RNases.

Antibodies that were obtained using *Escherichia coli*-expressed major coat protein p22 have too low titer in ELISA to be mass produced [4]. This can be potentially explained by the denatured state of expressed recombinant CP BYV during isolation on Ni-NTA agarose. It is highly likely that the 3D structure of denatured protein is significantly different from that of native protein when it is incorporated in the viral capsid. Thus, immunization with the isolated recombinant CP BYV leads to the rise of a large percentage of antibodies to an inner part of the viral capsid, that are useless, so the overall titer of antibodies is insufficient for ELISA testing. These technical challenges can be potentially overcome by using polypeptides that contain linear epitopes from ectodomains of viral CP and have high antigen index by Jameson–Wolf criteria [5]. However, this requires the knowledge of CPs’ 3D structure.

The aim of this project is to obtain antibodies to predict surface epitopes of BYV, using computer modeling of the 3D virion structure, and to test their suitability to be used for the BYV diagnostics in infected plants. For this purpose, we have employed the following methods:Computer modeling. The prediction of the 3D structure of BYV virion was performed using software SWISS-MODEL Workspace (Team @ Biozentrum Basel) [6,7]. As a most suitable template, the program used the already calculated structure of potexvirus (Pepino Mosaic Virus) CP, which served as a template to build a 3D model of BYV CP. The program also allows prediction of the disposition of CP subunits in assembled virion. So, it become possible to predict the external parts of CP accessible for antibodies recognition.Cloning of outer epitopes sequences of CP gene into the expression vector. To express epitopes, their nucleotide sequences were cloned into pQE40 plasmid in frame with the murine gene of dihydrofolate reductase (DHFR).Analysis of obtained sera using leaf extract from *Tetragonia tetragonioides* infected with BYV.

## 2. Materials and Methods

We used the Ukrainian isolate of BYV (GenBank: X73476), which has 99.4% nucleotide identity with other sequenced strains of this virus [8]. A cDNA copy of the capsid protein gene of the Ukrainian BYV isolate was used as a template for amplifying the epitope sequences. Commercial BYV positive control (PC-0981) and antibodies against BYV (AS-0185 IgG) were purchased from the German company DSMZ.

Amplification of epitope sequences was carried out in a Terzik MC2 amplifier (Pushchino, Russia). The epitope sequences were cloned in the Qiagen pQE40 expression vector, containing the mouse dihydrofolate reductase gene, using SibEnzyme (Novosiobirsk, Russia) restriction enzymes. The primers were synthesized at the SINTOL Company (Moscow, Russia).

### 2.1. Cloning Sequences of Predicted Epitopes of P22 BYV

To clone the nucleotide sequence of epitope one (**ep1**) N-GSAEPISAIATFENVSLADQTCLHGEDCDK-C into the expression vector pQE40 from Qiagen, the restriction endonuclease sites BglII and PstI were introduced in the primer sequence (underlined).

5′-tttagatctGGATCAGCTGAACCTATTAG-3′-(**ep1**-BYV+)

5′-tttctgcagtcactcttcgaagttcttcc-3′ (**ep1**-BYV-).

The vector was digested with BamHI-PstI endonucleases.

To clone the nucleotide sequence of the second epitope (**ep2**) N-GTSNKVNVQPTSTFIKASFGGGK-C, we used primers with the same restriction sites:

5′-tttagatctggcacttccaacaaagttaa-3′ (**ep2**-BYV+)

5′-tttctgcagttagtgagtgaggtacagttc-3′ (**ep2**-BYV-).

For protein expression, we used the M15 strain of *E. coli*. Cells were induced with 1 mM IPTG in 50 mL LB medium for 2 h at 37 °C. Analysis of the expression of target recombinant proteins DHFR-**ep1** and DHFR-**ep2** was performed by electrophoresis in 15% PAAG under denaturing conditions.

### 2.2. Isolation of the BYV Epitopes Fused with DHFR

Translated fused proteins were isolated by the affinity chromatography on the Ni-NTA column under denaturing conditions, strictly following the protocol of the Qiagen company. The most part of proteins was eluted with the buffer solutions D (100 mM NaH_2_PO_4_, 10 mM Tris·Cl, 8 M urea, pH to 5.9) and E (100 mM NaH_2_PO_4_, 10 mM Tris·Cl, 8 M urea, pH to 4.5) [9]. These fractions were then dialyzed against PBS. The protein concentration was determined spectrophotometrically, using the calculated extinction coefficients. 

### 2.3. Positive Controls for Testing Antisera

As a source of beet yellows virus, we used New Zealand spinach (*Tetragonia tetragonioides*) plants infected with BYV and kindly provided by E.M. Egorova.

For propagation, the virus was transferred from an infected plant to a fresh plant by grafting. After 30 days, the virus was isolated from the infected plants according to the following protocol: 200 g of symptomatic leaves were grinded in a mortar in two volumes of cold 0.5 M borate buffer (pH 8.0). Triton-X100 to 0.5%, β-ME to 0.05% were added and the cocktail was slowly stirred on a magnetic stirrer at 0 °C for 1 h. The debris was removed at 10,000 g for 10 min. The virus was precipitated from the supernatant on a Hitachi ultracentrifuge in an RP50t rotor at a speed of 100,000 g for 1 h through a 20% sucrose cushion. The pellet was dissolved in 0.01 M borate buffer and soluble fraction was precipitated by high-speed centrifugation again. Concentration of the purified virus was determined as described earlier for the potato leaf roll virus [10].

Another positive control for the antisera against the BYV was purchased from DSMZ company (Germany).

### 2.4. Electron Microscopy

For the electron microscopy, the virus preparation was stained with 2% uranyl acetate acidified with acetic acid. Transmission electron microscopy images were obtained by JEM-1011 microscope (JEOL Ltd., Tokyo, Japan)

### 2.5. Immunization of Laboratory Animals

Pre-immune sera were sampled from nine female outbred mice. Mice were subcutaneously immunized (3 mice for each antigen) three times with an interval of 7–10 days.

The first and next two immunizations were performed with 0.2 mg of **ep1**, **ep2** fused with DHFR preparations and a mixture of **ep1** and **ep2**. The first injection was with the addition of 30% complete Freund’s adjuvant, the second one—with the addition of 50% incomplete adjuvant. The third immunization was carried out without the addition of an adjuvant. Each time, immunization was performed by injections at 6 points.

### 2.6. Obtaining Antisera

Mice were bled one week after the third immunization. Immune sera were obtained by methods commonly used in the immunology [11]. Serum aliquots were frozen with liquid nitrogen and stored at −80 °C.

### 2.7. Antisera Testing

Sera and antisera were diluted with phosphate-saline buffer (PBS), pH 7.4.

Preliminary analysis of antisera against of the CP BYV was carried out by dot-ELISA as described below.

1 μL of analytes were immobilized on the Protran BA-85 (Schleicher&Shull) nitrocellulose membranes. Then membranes were washed in PBS, blocked with 5% BSA for 30 min at room temperature and probed with diluted (1:125, 1:625, 1:3125) antiserum in a shaker for 2 h at room temperature Unbound primary antibodies were washed out with PBS-0.1% BSA (3 × 5 min). Membranes with uploaded analytes were incubated for 2 h at room temperature in a solution of the goat anti-mouse secondary AuroProbe BL Plus (Amersham Pharmacia, Buckinghamshire, UK) antibodies diluted 100-fold with PBS, washed out with PBS-0.5% Tween-20 and stained with silver enhancement (BBI Solutions, Edinburgh, UK).

Quantitative characterization of the antisera against **ep1**, **ep2** and antiserum to the mixture of **ep1** and **ep2** specificity was determined by the solid phase indirect ELISA [12]. After each stage of ELISA, unbound substances were washed trice with PBS-0.05% Tween-20.

Fused DHFR-**ep1,** DHFR-**ep2,** a mixture of them and two positive controls were immobilized in triplicate on the Nunc MaxiSorp (Roskilde, Danmark) plates for 12 h at 4 °C. Then plates were blocked with 5% BSA for 20 min at 37 °C.

The 500-fold diluted antisera (optimal dilution was found by dot-ELISA) under study were added to the antigens, plates were incubated overnight at 4 °C. Ten thousand-times diluted with 0.5% BSA, the goat anti-mouse secondary antibodies conjugated with horse radish peroxidase were added and immune complexes were incubated for 2 h at room temperature. Finally, immune plates were washed, and the bound HRP-conjugate was detected with a peroxidase substrate 0.001 M H_2_O_2_ and ABTS (Biochemica AppliChem, Billingham, UK) in 0.002 M phosphate–citrate buffer, pH 4.4 at 37 °C for 20 min. Optical density of the reaction mixture was measured at 450 nm. To quantify the spread of a set of data, we used the sample standard deviation.

## 3. Results

### 3.1. Positive Controls

Plants infected with BYV had characteristic symptoms. Electron micrograph of isolated virus demonstrated the presence of filamentous virions approximately 1.5 micron in length, which is typical for BYV (Figure 1). Purified BYV preparation served as a positive control for the antisera specificity examination.

### 3.2. Cloning of BYV Epitopes

The scheme describing the cloning of polynucleotides encoding the epitopes fused to the DHFR gene is presented in Figure 2.

We used optimized codons for strong protein expression in *E. coli* [13], see Appendix A.

### 3.3. Selection of Epitopes Using 3D-Model of BYV p22 Coat Protein

The proposed 3D model of BYV CP (Figure 3) was built using SWISS-MODEL Workspace software (version 8.05) Team @ Biozentrum Basel/SIB, which can be found on server https://swissmodel.expasy.org/interactive. As a template for building the 3D structure of BYV CP, the software has selected the 3D structure of Pepino Mosaic Virus (5FN1), a member of the potexvirus family, which, similar to closteroviruses, has a flexible filamentous virion structure.

By replacing in the known 3D structure of potexvirus virion capsid protein, the coat protein of closterovirus, we obtained the proposed 3D structure of BYV virion that is formed by the BYV p22 protein (Figure 4).

As Figure 4 shows, epitopes **ep1** and **ep2** are located on the surface of the proposed 3D structure of the BYV virion fragment, and thus may potentially serve as antigens that can induce the immune response.

The isolation of BYV epitopes fused to DHFR was performed using affinity chromatography (see Methods). The majority of the epitope–DHFR proteins was eluted in fractions E1–E2 (Figure 5), which then were dialyzed against phosphate buffer and used for the mice immunization.

By immunizing mice with epitopes fused with murine DHFR, we obtained a pool of antibodies that specifically recognize the epitopes (Figure 6 and Figure 7).

### 3.4. Antiserum Analyses

#### 3.4.1. Dot-ELISA

The initial analysis of antisera obtained to epitopes of CP BYV was performed using dot-ELISA with chromogenic detection (Figure 6A–C). The sensitivity of obtained antiserum to **ep1** with a chromogenic substrate was found to be at least 10 ng of virus in 1 mL of plant extract, demonstrating high specificity and effectiveness at 500-fold dilution.

#### 3.4.2. Plate Indirect ELISA

Quantitative sensitivity of antisera to the purified fused **ep1**, **ep2** as well as their mix was determined using solid phase ELISA (Figure 7) in the indirect format [13].

We found that purified BYV and in the extract of the infected plant can be detected at concentration about 4 ng/mL even with a chromogenic substrate by antisera to **ep1** and **ep2** at 1:500 dilution. Furthermore, the titration curves indicate that antiserum to **ep1** is more efficient in the detection of native BYV than antiserum to **ep2**, which is in line with the proposed 3D structure where epitope 1 is located closer to protein surface (Figure 4) and thus may be more readily accessible to antibodies (Figure 7A–C).

## 4. Discussion

The low molecular weight of epitope peptides have low immunogenicity [14,15]. To improve the immunogenicity of the epitopes of interest, we used a bacterial expression vector pQE40 containing a gene encoding murine DHFR. We cloned fused polynucleotide sequences encoding the epitopes in frame with the DHFR gene (Figure 1). Expressed recombinant proteins (Figure 2), which contained 6-His tag, were purified using affinity chromatography on the Ni-NTA column under denaturing conditions. We have initially assumed that the conformation of low molecular weight epitopes in these denatured recombinant proteins are close to the conformation of low molecular weight epitopes in aqueous solutions. This was later confirmed by the results of antisera immunochemical analysis and proposed by us 3D models of CP BYV and a fragment of the BYV virion.

*In silico* models for predicting 3D structures of proteins [5,6,7,16] and viral capsid [17,18] have been successfully adopted in the last decades. Homology modelling has proven to be the approach that generates a reliable 3D model of a protein from its amino acid sequence. There are several computational programs for the protein structure determination [16,17]. We chose the SWISS-MODEL Workspace service [6,7].

In the proposed models of p22 protein (Figure 3) and the BYV virion fragment (Figure 4), the N-terminus and C-terminus of the closterovirus CP protein are exposed on the surface of the viral particle and thus may be potentially immunogenic. Moreover, fusing them to a murine full-size protein should increase their immunogenicity even further.

By immunizing mice with epitopes fused to murine DHFR protein, we were able to obtain a pool of the antibodies that can specifically recognize the epitopes (Figure 6 and Figure 7). Absence of the potential contamination with plant proteins, which can produce high background in ELISA, is an additional benefit of using recombinant bacterial proteins for immunization.

Antibodies raised against the epitopes that we have selected, based on a 3D model of CP BYV, were demonstrated to recognize positive controls (Figure 6 and Figure 7) with a sensitivity that allows a practical use of these antibodies in diagnostics.

We are continuing to develop novel approaches from the bio-safe method for the RNA isolation [19] from viruses, bacteria, plants and animal cells, to the immune detection of the viroid RNA [20] for the large-scale molecular diagnostics of viral infections, with the intention to use them in practice. For the detection of hard-to-isolate viruses, the last method relies on a computer modeling of the coat protein’s and virion fragment’s 3D structures (current work).

## 5. Conclusions

Thus, to diagnose an infection by hard-to-isolate or dangerous-for-human-health viruses, which antigens are difficult to obtain by immunochemical methods, it is not necessary to purify the actual virus. One can just employ the polynucleotide sequence, which encodes viral coat proteins. Currently, when the sequencing of a whole genome can be completed in as fast as 2–3 days, detection and identification of an exogenous viral genome in the infected host does not present a challenge. Using bioinformatics to perform a search for a virus with similar genome, one can build the 3D model of the virus CP and the virus particle by comparison (as a whole or in part). The immunogenicity of the virus CPs’ epitopes can be determined following Jameson–Wolf criteria.

The suitability of the isolated antisera (antibodies), from healthy and infected organisms, for practical use can be determined by immunochemical analysis. The proposed approach is significantly less labor-intensive than the one that relies on the isolation and purification of dangerous or hard-to-isolate virus.

## Figures and Tables

**Figure 1 biotech-11-00052-f001:**
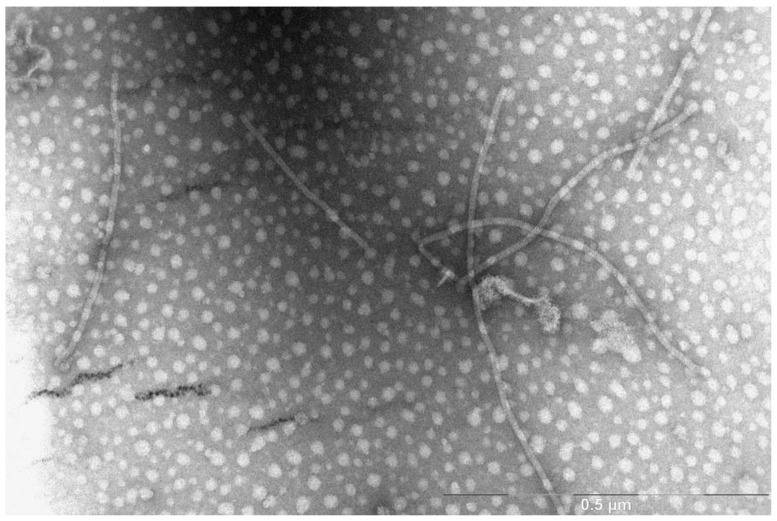
Electron micrograph of BYV isolated from *T. tetragonioides*.

**Figure 2 biotech-11-00052-f002:**
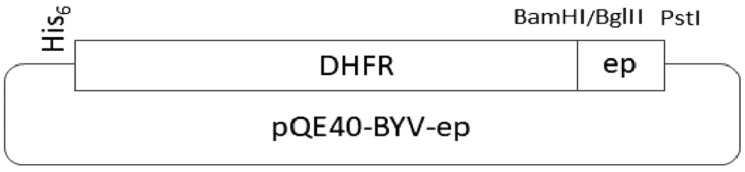
Scheme of the vector that expresses the epitope.

**Figure 3 biotech-11-00052-f003:**
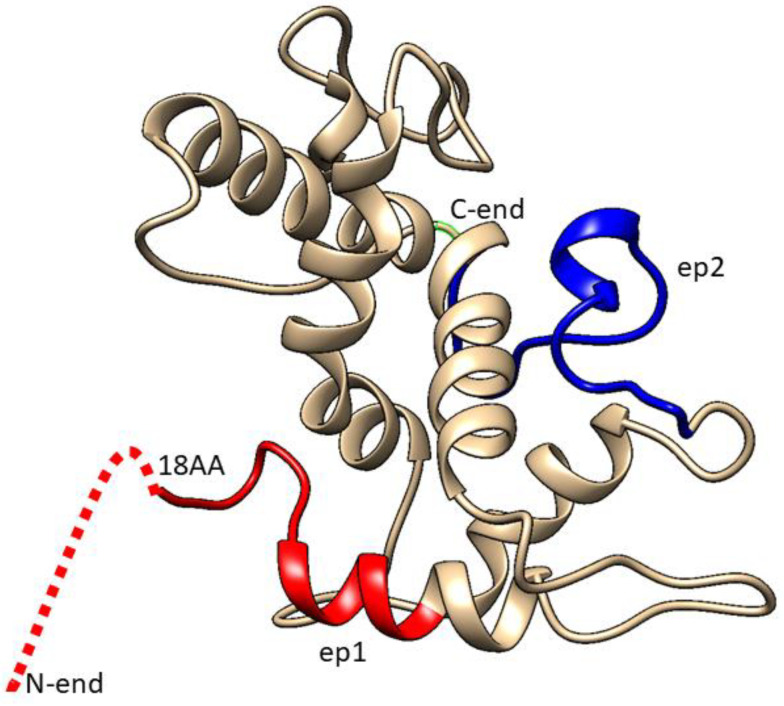
Three-dimensional model of surface protein p22. Red color designates **ep1**, blue color designates **ep2**, while dashed line marks that part of the epitope whose structure could not be determined.

**Figure 4 biotech-11-00052-f004:**
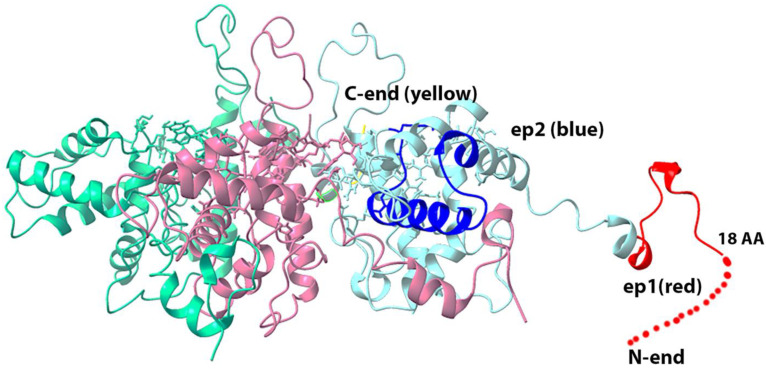
Three-dimensional model of the BYV virion fragment, based on the Pepino Mosaic Virus structure (5fn1.pdb rcsb.org). Red color designates **ep1**, deep blue color designates **ep2**, yellow color designates C-terminus of CP that is located inside the virion, while dashed line marks the disordered part of the epitope, whose structure could not be determined.

**Figure 5 biotech-11-00052-f005:**
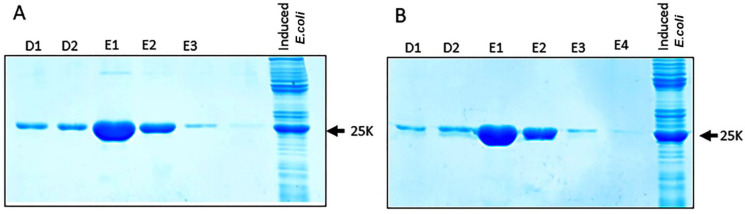
Electrophoretic analysis in 15% polyacrylamide gel of the protein fractions eluted from the Ni-NTA column: (**A**) fused DHFR-**ep1** protein; (**B**) fused DHFR-**ep2** protein. The outermost right lane demonstrates the level of epitopes’ expression after the induction relative to total protein in *Escherichia coli.* D1-D2 и E1-E4 designate 0.5 mL successive fractions elution with D and E buffer solutions (see Materials and Methods).

**Figure 6 biotech-11-00052-f006:**
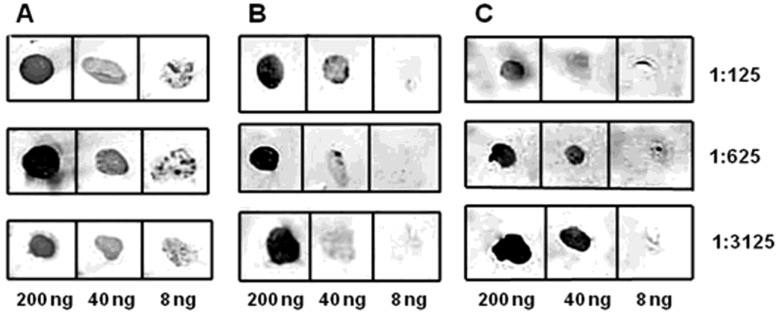
Dot-ELISA of the murine antisera obtained. The dilution ratios for antisera are indicated on the right, while the quantity of used CP BYV epitope is indicated at the bottom of the figure. (**A**) antiserum to **ep1**; (**B**) antiserum to **ep2**; (**C**) antiserum to the mix of **ep1** and **ep2**.

**Figure 7 biotech-11-00052-f007:**
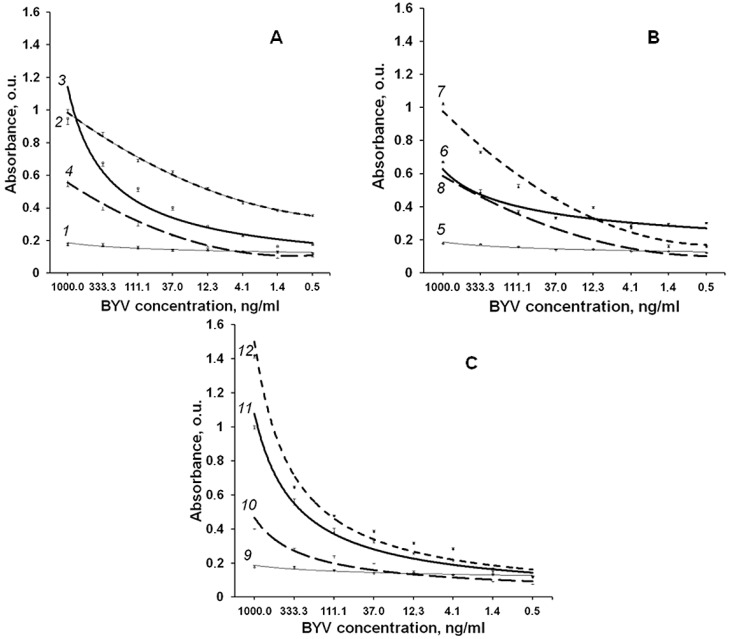
Sensitivity of the obtained antisera. (**A**) antiserum to **ep1**; (**B**) antiserum to **ep2**; (**C**) antiserum obtained to mix of the epitopes **ep1** and **ep2**. Curves’ numbers show antiserum-antigen pairs analysed: 1, 5, 9—pre-immune sera and native BYV; 2—antiserum to **ep1** and **ep1**; 3—antiserum to **ep1** and native BYV; 4—antiserum to **ep1** and healthy plant control; 6—antiserum to **ep2** and native BYV; 7 antiserum to **ep2** and **ep2**; 8—antiserum to **ep2** and healthy plant control; 10—antiserum obtained to the mix of epitopes **ep1**, **ep2** and healthy plant control; 11—antiserum obtained to the mix of epitopes **ep1**, **ep2** and native BYV; 12—antiserum obtained to the mix of epitopes **ep1**, **ep2** and mix of these epitopes.

## Data Availability

Not applicable.

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
