# Peer review of "Bioinformatics Predicted Linear Epitopes of the Major Coat Protein of the Beet Yellows Virus for Detection of the Virus in the Cell Extract of the Infected Plant"

_biotech, 2022, doi:10.3390/biotech11040052_

Round 1

Reviewer 1 Report

In present study, the authors discover epitopes of beet yellows virus (BYV) major coat protein p22 using SWISS-MODEL Workspace (Biozentrum Basel) protein 3d prediction program. The authors showed that murine antibodies were raised against each epitope and combination of both, and characterized by dot-ELISA and indirect ELISA. Andmore, the authors used these antibodies for diagnosis of virus disease in systemically infected Tetragonia tetragonioides. However, several points should be addressed before the MS considered for publication in BioTech:

1)        In this study, pepino mosaic virus coat protein was served as a template to build a 3d-model of BYV CP, and predict the external parts of CP accessible for antibodies recognition. How to confirm the specificity of antibody? How about the sensitivity and specificity of antibodies produced by this method, especial to other virus species in genus Closterovirus?

2)        Line 9, “genus Closterovirus, family Closteroviridae” should be “o genus Closterovirus, family Closteroviridae”.

3)        Line 15 and 42, “E. coli” should be “Escherichia coli”.

4)        Line 25, “capsid” should be “capsid”. And removed the short red line.

5)        Many editorial errors like above were also found in the MS and references. The authors should rewrite this MS carefully.

Author Response

            Thank you for your careful review of our manuscript entitled “Bioinformatics Predicted         Linear Epitopes of the Major Coat Protein of the Beet Yellows Virus for Detection of the Virus in            the Cell Extract of the Infected Plant”. I think the manuscript has been improved after the recommended corrections.

Point 1. In this study, pepino mosaic virus coat protein was served as a template to build a 3d-model of BYV CP, and predict the external parts of CP accessible for antibodies recognition. How to confirm the specificity of antibody? How about the sensitivity and specificity of antibodies produced by this method, especial to other virus species in genus Closterovirus?

Response 1. Thank you.

We tried to prove the antisera specificity using two independent controls. First, analyzing plant infected with the BYV. We obtained the virus preparation and examined it with the transmission electron microscopy and by ELISA (Figure 1). The virus preparation obtained was readily recognized as by antisera raised against epitopes ep1, ep2 and ep1+ep2 so and by commercial antibodies (DSMZ Germany).

However, we did not use other Closterovirus species.

                Point 2. Line 9, “genus Closterovirus, family Closteroviridae” should be “genus Closterovirus,             family Closteroviridae”.

            Thanks. Done. Line 10.

            Point 3. Line 15 and 42, “E. coli” should be “Escherichia coli”.

            Done. Lines15 and 40.

            Point 4. Line 25, “capsid” should be “capsid”. And removed the short red line.

            Done. Line 25.

            Point 5. Many editorial errors like above were also found in the MS and references. The authors   should rewrite this MS carefully.

            Thank you. We regret about it and tried to do our best.

Reviewer 2 Report

Plant viruses are important threats to food security and there is a need for novel approaches and tools for their rapid detection. Skurat et al. describe a method for the rapid development of an immunoassay to detect beet yellows virus (BYV).

The results are interesting and within the scope of the journal; however, there are some aspects that should be considered to enhance the manuscript:

- L28, to better contextualize the findings I would suggest to include a reference that describes closteroviruses and their genetic properties (e.g. Virus Res. 2006 Apr;117(1):38-51. doi: 10.1016/j.virusres.2006.02.002)

- L70, include GenBank accession numbers or references of the "other sequenced strains of this virus"

- The authors mention that epitope sequences were amplified from viral cDNA (L71-72), but then say "We used optimized codons" (L78). Can they clarify which nucleotide sequences were used in their E. coli expression constructs? I would recommend to make the complete sequences of pQE40-BYV-ep1 and pQE40-BYV-ep2 available within the supporting information.

- L96, "eluted in E1-E2" is unclear

- L101-102, to enhance clarity I would replace the sentence with "Tetragonia tetragonioides (kindly provided by E.M. Egorova), commonly called New Zealand spinach, was used for propagation of BYV."

- L108, replace "10.000 rpm" with the g value to be consistent with the rest of the manuscript

- L115, please include DSMZ product number

- L154, write "at r.t." in full

- Fig. 4 has poor resolution, I would suggest to enhance or remove it. Additionally, can the authors provide any statistics supporting the confidence of the model?

- I would suggest to split Fig. 5 in panels A and B to separately show the gels and move the extra description text to the caption (as done in Fig. 6)

- Fig. 6, should the authors present results of the "pre-immune sera" and healthy plant controls?

- Fig. 7, should the authors present results of the healthy plant controls?

- Antibodies and immunoassays for BYV detection are commercially available, for instance from DSMZ (product numbers AS-0185 and RT-0185 at www.dsmz.de). Can the authors provide any comparison of the sensitivity of their antibodies and those commercially available?

- Some abbreviations are unclear RF (L73), RUS (75), etc.

Author Response

Point 1. L28, to better contextualize the findings I would suggest to include a reference that describes closteroviruses and their genetic properties (e.g. Virus Res. 2006 Apr;117(1):38-51. doi: 10.1016/j.virusres.2006.02.002)

Response 1. Thank you. We put it in the reference list (references 1 and 2.

Point 2. L70, include GenBank accession numbers or references of the "other sequenced strains of this virus"

Response 2. Thanks. Closterovirus genome sequence data were insert in the reference list (Reference 8).

Point 3. The authors mention that epitope sequences were amplified from viral cDNA (L71-72), but then say "We used optimized codons" (L78). Can they clarify which nucleotide sequences were used in their E. coli expression constructs? I would recommend to make the complete sequences of pQE40-BYV-ep1 and pQE40-BYV-ep2 available within the supporting information.

Response 3. Thank you for this remark. We added response to the Supplement section.

Point 4. L96, "eluted in E1-E2" is unclear

Response 4. Thank you. We clarified it by replacement with a description of the buffer solutions (D and E) used (lines 90,91).

Point 5. L101-102, to enhance clarity I would replace the sentence with "Tetragonia tetragonioides (kindly provided by E.M. Egorova), commonly called New Zealand spinach, was used for propagation of BYV."

Response 5. Thanks. Done (line 96).

Point 6. L108, replace "10.000 rpm" with the g value to be consistent with the rest of the manuscript

Response 6. Thanks. Done (line 101).

Point 7. L115, please include DSMZ product number

Response 7. Done (line 63, 64).

Point 8. L154, write "at r.t." in full

Response 8. Done. (line 142).

Point 9. Fig. 4 has poor resolution, I would suggest to enhance or remove it. Additionally, can the authors provide any statistics supporting the confidence of the model?

Response 9. Thank you. We replaced this figure with its fragment. Thus, the resolution has increased.

Point 10. I would suggest to split Fig. 5 in panels A and B to separately show the gels and move the extra description text to the caption (as done in Fig. 6)

Response 10. We did it as you recommended (Fig. 5).

Point 11. Fig. 6, should the authors present results of the "pre-immune sera" and healthy plant controls?

Response 11. Please find it in Figure 7 and its legend.

Point 12. Fig. 7, should the authors present results of the healthy plant controls?

Response 12. Yes, see Figure 7.

Point 13. Antibodies and immunoassays for BYV detection are commercially available, for instance from DSMZ (product numbers AS-0185 and RT-0185 at www.dsmz.de). Can the authors provide any comparison of the sensitivity of their antibodies and those commercially available?

Response 13. Antisera to epitopes are weaker than commercial antibodies. Specially, we compared sensitivity of the BYV detection with commercial antibodies, antibodies to the irreversibly denatured p22 protein and   to epitopes ep1, ep2 and ep1+ep2 (manuscript entitled

Comparative Immunodiagnostics of   the Beet Yellows Virus Infections with Antisera against Predicted Epitopes, Irreversibly Denatured Coat Protein and  Commercial Antibodies to the Virus by Konstantin O. Butenko, Eugene V. Skurat, Аlex M. Arutyunyan , Yuri F. Drygin∆,* is in preparation)

Point 14. Some abbreviations are unclear RF (L73), RUS (75), etc.

Response 14. Done. Both mean Russian Federation.

Thank you for your careful review of our manuscript entitled “Bioinformatics Predicted Linear Epitopes of the Major Coat Protein of the Beet Yellows Virus for Detection of the Virus in the Cell Extract of the Infected Plant”. I think the manuscript has been significantly improved after the recommended corrections.

Round 2

Reviewer 1 Report

Mostly my concens have been addressed. If possible, I think the authors should add  the sensitivity and specificity of antibodies to other virus species in genus Closterovirus?

Author Response

Point 1. Mostly my concens have been addressed. If possible, I think the authors should add  the sensitivity and specificity of antibodies to other virus species in genus Closterovirus?

Response 1. Dear Reviewer,

Thank you for this interesting question. I should say that we are mostly dealing with potato viruses. Since 1996, we collaborate with the Lorch Potato Research Institute, and later with the Research and Development firm “Immunotek” at Lomonosov MSU to accommodate modern molecular diagnostics of potato virus infections for practical use. Please find some our reference publications:

Yu.F. Drygin, A.N. Blintsov, V.G. Grigorenko, et al., Highly sensitive field test lateral flow immunodiagnostics of PVX infection. Appl Microbiol Biotechnol (2012) 93:179–189; .DOI 10.1007/s00253-011-3522-x and references 3, 11, 20 of the current manuscript.

Beet yellows virus appeared as our target because previous attempts of our colleague (Dr. Agranovsky, personal communication) to obtain good antibodies against the BYV recombinant p22 for detection virus in the infected beet showed limited abilities of using them in practice. Because BYV is hard to access virus, we proposed another approach to resolve the problem. We described it in the manuscript under your review. BYV was the only closterovirus available to us.

Regarding your inquiry, we contacted with our colleagues from the Department of Virology to find another than BYV closterovirus. No luck. It might be that you work with closteroviruses. Will you be interesting in cooperation to get answer for your question, please let me know? We will try to do our best.

Reviewer 2 Report

I would like to thank to the authors for addressing most of the comments of the previous review round.

I have just a couple of recommendations to enhance the manuscript clarity:
L61, I think "99.4% genome homology" should be replaced with "99.4% nucleotide identity" (for the correct use of homology see for instance Cell. 1987 Aug 28;50(5):667. doi: 10.1016/0092-8674(87)90322-9)
L62, the link provided in reference 8 is continuously updated to include any newly released sequence, further partial genomic sequences are listed making the methods unclear. Please include the complete list of the used accession numbers e.g. GenBank: NC_001598.1, MT701720.1, MT815988.1, etc.
L110, please include the manufacturer and model of the transmission electron microscope used.

Author Response

Point 1. L61, I think "99.4% genome homology" should be replaced with "99.4% nucleotide identity" (for the correct use of homology see for instance Cell. 1987 Aug 28;50(5):667. doi: 10.1016/0092-8674(87)90322-9)

Response 1. Thank you for this subtle comment on a terminology muddle. We followed your recommendation. Line 61.

Point 2. L62, the link provided in reference 8 is continuously updated to include any newly released sequence, further partial genomic sequences are listed making the methods unclear. Please include the complete list of the used accession numbers e.g. GenBank: NC_001598.1, MT701720.1, MT815988.1, etc.

Response 2. Thank you for this critical comment. We added current GenBank accession numbers of the BYV genome: ON738341.1  MT815988.1 MT701720.1 X73476.1 AF056575.1 OL472076.1 in the REFERENCES. Line 302.

Point 3. L110, please include the manufacturer and model of the transmission electron microscope used.

Response 3. Done. Transmission electron microscopy images were obtained by JEM-1011 microscope (JEOL Ltd., japan). Line 111.
